# Analysis of Potential Biomarkers in Frontal Temporal Dementia: A Bioinformatics Approach

**DOI:** 10.3390/ijms241914910

**Published:** 2023-10-05

**Authors:** Inara Deedar Momin, Jessica Rigler, Kumaraswamy Naidu Chitrala

**Affiliations:** Department of Engineering Technology, University of Houston, Sugar Land, TX 77479, USA; idmomin@uh.edu (I.D.M.); jdrigler@uh.edu (J.R.)

**Keywords:** frontal temporal dementia, comorbidity, breast cancer, COVID-19, differential gene expression analysis

## Abstract

Frontal temporal dementia (FTD) is a neurological disorder known to have fewer therapeutic options. So far, only a few biomarkers are available for FTD that can be used as potential comorbidity targets. For example, genes such as *VCP*, which has a role in breast cancer, and *WFS1*, which has a role in COVID-19, are known to show a role in FTD as well. To this end, in the present study, we aim to identify potential biomarkers or susceptible genes for FTD that show comorbidities with diseases such as COVID-19 and breast cancer. A dataset from Gene Expression Omnibus containing FTD expression profiles from African American and white ethnicity backgrounds was included in our study. In FTD samples of the GSE193391 dataset, we identified 305 DEGs, with 168 genes being up-regulated and 137 genes being down-regulated. We conducted a comorbidity analysis for COVID-19 and breast cancer, followed by an analysis of potential drug interactions, pathogenicity, analysis of genetic variants, and functional enrichment analysis. Our results showed that the genes *AKT3*, *GFAP*, *ADCYAP1R1*, *VDAC1*, and *C4A* have significant transcriptomic alterations in FTD along with the comorbidity status with COVID-19 and breast cancer. Functional pathway analysis revealed that these comorbid genes were significantly enriched in the pathways such as glioma, JAK/STAT signaling, systematic lupus erythematosus, neurodegeneration-multiple diseases, and neuroactive ligand–receptor interaction. Overall, from these results, we concluded that these genes could be recommended as potential therapeutic targets for the treatment of comorbidities (breast cancer and COVID-19) in patients with FTD.

## 1. Introduction

Frontotemporal disorder or frontal temporal dementia (FTD) is a disease with symptoms such as impaired thinking, reasoning, memory, and other functions leading to interferences in performing daily activities, causing diverse brain degeneration [1,2]. In the USA, FTD is a prevalent neurodegenerative dementia with the lowest survival rate of only 3–14 years after the onset and is known to be most prevalent among individuals aged ≤65 years. To date, there are no effective therapies available for treating FTD or decreasing its progression [3].

Currently, there are several factors that confer susceptibility risk to FTD. Among them, genetic risk factors such as non-coding hexanucleotide repeat expansion (HRE) located in the *C9orf72* gene known to be prevalent [4,5]. Other risk factors include mutations in the genes such as *GRN* and *MAPT*. Several new therapeutic strategies targeting these genes are under development, and some of them are moving into clinical trials [6]. Though clinical subtypes and genetic factors that confer risk for FTD have been recently identified, molecular mechanisms driving its pathogenesis are not much understood. In recent years, transcriptomic studies and network biology methods targeting the disrupted pathways have been instrumental in understanding the molecular mechanisms in neurodegenerative disorders.

FTD patients show a significantly lower prevalence of cancer [7] and exhibit reciprocal phenomena with gene *C9orf72*, which has non-coding hexanucleotide repeat expansion that causes FTD genetically. Hexanucleotide repeat expansions of more than 10 repeats located in the gene *C9orf72* were found to be a risk for high pathogenicity of COVID-19 phenotypes, indicating a shared genetic profile between FTD and COVID-19. The molecular mechanisms driving such behavior or the genes conferring this comorbidity are unknown, specifically among the racial/ethnic groups such as the African Americans (AA) and white populations. A recent study on gene co-expression network analysis of FTD-related genes in the frontal and temporal cortices of FTD patients showed enrichment of pathways related to DNA metabolism, transcriptional regulation, and DNA protection. These results indicate that network-based approaches could offer valuable insights into the pathogenesis of FTD, thereby aiding the identification of potential treatment targets [8]. Therefore, in the current study, we aim to utilize gene expression data from the public domain databases to identify differentially expressed genes (DEGs), followed by the identification of their comorbidity status in breast cancer and COVID-19 using bioinformatics approaches.

## 2. Results

### 2.1. Differential Expression Analysis of FTD Genes

The flowchart of the overall data analysis performed in our study is shown in Figure 1. Differential expression analysis between the control (n = 8) and FTD (n = 8) samples from African American (AA) and white populations (Table 1) from the GSE193391 dataset showed that 305 DEGs were found to be significant at a *p*-value < 0.05 with 168 genes being up-regulated and 137 genes being down-regulated (Appendix A). Among these DEGs, genes *RAB3A* (1.91), *NEFL* (1.89), *KCNV1* (1.88), *BDNF* (1.86), *YWHAH* (1.66), *GABRG2* (1.62), *SH3GL2* (1.58), *SYN2* (1.57), *RIMS1* (1.56), and *PAK1* (1.53) were found to be significantly up-regulated, and the genes *GFAP* (−3.08), *C4A/B* (−2.71), *CD44* (−2.36), *ENTPD2* (−2.27), *CD163* (−1.94), *PLXNB3* (−1.76), *IL18* (−1.65), *SPARC* (−1.60), *NWD1* (−1.60), and *GSN* (−1.59) were found to be significantly down-regulated based on the log2fold change value. These significant DEGs were represented using the box plot (Appendix A) and shown in the volcano plots (Appendix A).

### 2.2. Analysis of DEGs Involved in Autophagy

Results from human autophagy database analysis showed that among the DEGs that are significantly expressed across AA and white populations, *MAPK9* was found to be involved in autophagy (Figure 1), indicating that this gene is essential for survival, differentiation, development, and homeostasis in FTD. Further, results from the Variation effect on Protein Structure and function (VAPROS) [9] database analysis on the Autophagy database showed that the genes *AKT3*, *GFAP*, *UBE2N*, and *VDAC1* were also found to be involved in autophagy (Figure 1). Gene *AKT3* showed a cluster of 540 genes located in the cytoplasm, nucleus, and cell membrane. Gene *GFAP* showed a cluster of 675 genes located in the cytoplasm. Gene *UBE2N* showed a cluster of 483 genes in an unknown location. Gene *VDAC1* showed a cluster of 205 genes located in the mitochondrion outer membrane. Among these identified Autophagy genes, *GFAP* is treated as a measure for astrogliosis—a known pathological process in FTD and *GFAP*, whose levels are found to be associated with the intensity, progression, and survival of the disease. Furthermore, *GFAP* reduction is known to be associated with the *WFS1* gene, which is found to be reduced in the FTD samples.

### 2.3. Analysis of the Genes Involved in COVID-19

For the genes that were found to be involved in autophagy, we performed DisGeNET analysis (Appendix A) to identify their respective association with COVID-19. Results from the DPI analysis showed that the genes *GFAP* and *RTN4* were found to be involved in COVID-19 with a score of 0.885 and 0.692. DSI analysis showed that the gene *GFAP* had a DSI value of 0.421, and the gene *RTN4* had a DSI value of 0.621. Furthermore, gene *GFAP* had a pLI value of 0.00000146, and gene *RTN4* had a pLI value of 0.45318. Both the genes *GFAP* and *RTN4* were found to be tolerable to protein truncating variation.

### 2.4. Evaluation of FTD DEG Expression in Breast Invasive Carcinoma and Survival Analysis

We then evaluated the expression of FTD DEGs in breast invasive carcinoma and non-cancer tissues. Results demonstrated that the gene *AKT3* (0.88), which is significantly up-regulated in FTD, is significantly down-regulated in breast invasive carcinoma patients (Figure 2A), whereas the genes *UBE2N* (0.766) and *VDAC1* (0.96) were significantly up-regulated in FTD are significantly up-regulated in breast invasive carcinoma patients also (Figure 2A). Further, the genes *ADCYAP1R1* (−1.166) and *GFAP* (−3.088) were significantly down-regulated in FTD and were also down-regulated in breast invasive carcinoma (Figure 2B), whereas *C4A* (−2.71), which is significantly down-regulated in FTD, is significantly up-regulated (Figure 2B) and expressed in breast invasive carcinoma patients compared to the normal controls at a *p*-value < 0.05. Overall, these results indicate that these genes play a significant role in breast cancer. Results from the survival analysis showed that the up-regulated genes *AKT3* and *VDAC1* and the down-regulated genes *ADCYAP1R1*, *GFAP*, and *C4A* were found to be significantly associated with the overall survival of the patients with breast invasive carcinoma (Figure 3). The overall heat map of FTD-Breast cancer comorbid gene expression in metastatic, solid tissue of normal and primary tumors of the TCGA datasets is shown in Figure 4. Further, we evaluated the immunotherapeutic drug interactions for these FTD-breast cancer comorbid genes as the immunotherapy is effective both in advanced and early setting phase 3 clinical trials of Breast cancer. Results from immunotherapeutic drug interactions analysis showed that *AKT3* has interactions with the 22 drugs (Appendix A). *GFAP* has interactions with 25 drugs (Appendix A). *VDAC1* and *ADCYAP1R1*, on the other hand, had interactions with three drugs (Appendix A). Additionally, we evaluated the list of RNA-binding proteins (RBPs) interacting with these FTD-Breast cancer comorbid genes. Results showed that there are 11 RBPs (EIF4A3, FBL, FMR1, IGF2BP1, IGF2BP2, IGF2BP3, MOV10, NOP56, NOP58, RBM47, SRSF1) interacting with *AKT3*, 2 RBPs (EIF4A3, IGF2BP2) interacting with *VDAC1*, 1 RBP (DGCR8) interacting with *ADCYAP1R1*, 2 RBPs (DGCR8, FUS) interacting with *GFAP*, and 27 RBPs (AUH, BCCIP, BUD13, CSTF2T, EIF4A3, FAM120A, GTF2F1, HNRNPA1, HNRNPC, HNRNPK, HNRNPUL1, IGF2BP1, IGF2BP3, LARP7, LIN28B, PRPF8, RBFOX2, SF3A3, SF3B4, SLTM, SMNDC1, SND1, SRSF1, SRSF7, SRSF9, TRA2A, U2AF1) interacting with *C4A* (Appendix A). We then conducted an analysis of respective gene interactions with RNA. Our results showed that *AKT3* has the highest, whereas *GFAP* has the lowest interaction for RBP (Appendix A). We have also conducted mRNA–RNA interaction analysis for the FTD-Breast comorbid genes. Results showed that *VDAC1* had the highest interaction, and the genes *ADCYAP1R1* and *GFAP* had the lowest interaction among the five genes (Appendix A).

### 2.5. Pathogenicity Analysis of FTD-Breast Cancer Comorbid Genes

We analyzed the pathogenicity for the FTD-Breast cancer comorbid genes (Appendix A). Results showed that the tools CADD, DANN, ReMM, and fitCons expressed the most “likely pathogenic” pathogenicity scores for the FTD-Breast comorbid genes. Tool fitCons showed that all five genes were “likely pathogenic”. DANN showed that the genes *AKT3*, *VDAC1*, *GFAP*, and *C4A* were “likely pathogenic”. CADD and ReMM tools showed that the genes *VDAC1*, *GFAP*, *C4A*, and *ADCYAP1R1* were “likely pathogenic”. LINSIGHT, GenoCanyon, Eigen, Eigen_PC, regBase PAT, and FATHMM-MKL tools showed that one or two genes were “likely pathogenic”. The CDTS tool did not show any of the five genes as “likely pathogenic” (Appendix A).

### 2.6. Genetic Alteration Analysis in FTD-Breast Cancer Comorbid Genes

For the five FTD-Breast cancer comorbid genes (*AKT3*, *GFAP*, *C4A*, *VDAC1*, *ADCYAP1R1*), we analyzed the alterations in the genetic profiles among the different subtypes of Breast cancer. Results showed that the *AKT3* gene has the highest alterations in the genetic biomarker at 11%, while *VDAC1* has the lowest alterations at 0.8%. *ADCYAP1R1* (3%), *GFAP* (2.5%), and *C4A* (2.3%) have similar percentages of genetic alterations. In all the genes, many of the alterations are mostly composed of amplification and deep deletion (Figure 5). To analyze these genes at the single-cell level, we performed the t-sne analysis in the Breast cancer subtypes, Triple Negative A (TNA), Triple Negative B (TNB), Luminal A (LA), Luminal B (LB), and Basal-like (BL) and H-subtype. Results showed that the gene *VDAC1* is highly expressed in all the subtypes, whereas *AKT3* is expressed in TNA, TNB, and H-subtypes (Appendix A), indicating that these genes have a significant role in Breast cancer at the single-cell level.

### 2.7. Clinical Association of FTD-Breast Comorbid Genes

We investigated whether common genetic variants of *AKT3* and *GFAP* genes have any direct relationship or clinical association with the pathology of FTD by identifying 3′aQTR on 3′aQTL-atlas (https://wlcb.oit.uci.edu/3aQTLatlas/index.php,) (accessed on 16 December 2022). Data of 3′aQTL variants from brain frontal cortex tissues for genes *AKT3* and *GFAP* was collected (Appendix A). Using the Gene/SNP search query on the brain frontal cortex tissue, the *AKT3* gene yielded 11 significant variants with single nucleotide variants (rs2502342, rs2998662, rs2998661, rs12077950, rs9725721, rs6429391, rs6429389, rs12075066, rs12078540, rs12073551, rs12087532) (Appendix A). None of the transcript variants showed that the *AKT3* gene posed a significant association with FTD [10]. The *GFAP* gene yielded 10 significant variants (rs12941832, rs34902223, rs9895349, rs4793148, rs3764840, rs3760382, rs4426386, rs2337848, rs9893320, rs9911454) in the human brain frontal cortex tissue (Appendix A). The genetic consequence of several of these 3′UTR variants affects the promoter region of the gene and encodes the candidate tumor suppressor gene, *ADAM11*. *ADAM11* is a highly conserved gene involved in biological pathways, including fertilization, muscle development, and neurogenesis (https://www.ncbi.nlm.nih.gov/gene/4185) (accessed on 20 November 2022). The *ADAM11* gene was found to be associated with the molecular pathology of Breast cancer, epilepsy, and familial frontotemporal lobe [11].

### 2.8. Pathway Enrichment Analysis

We further explored the interaction between the FTD-Breast comorbid genes (*AKT3*, *GFAP*, *C4A*, *VDAC1*, *ADCYAP1R1*) and their different aspects of biological pathways in relation to neurological changes by finding the pathways on https://www.kegg.jp/kegg/pathway.html (accessed on 6 January 2023) [12]. Results showed that the *AKT3* gene is enriched with multiple pathways of glial cells impacting the brain (Appendix A), such as cytokine–cytokine receptor interaction, ErbB signaling, mTOR signaling, calcium signaling, MAPK signaling, p53 signaling, cell cycle, cell growth, and cell proliferation. *GFAP* was enriched with the JAK/STAT signaling pathway (Appendix A), *C4A* was enriched with the pathways such as JAK/STAT signaling, cytokine-cytokine receptor interaction, B cell/T cell receptor signaling (Appendix A), *VDAC1* was enriched with neurodegeneration pathways of multiple diseases such as Alzheimer’s, Parkinson’s, Amyotrophic lateral sclerosis, Huntington’s, Spinocerebellar ataxia, and Prion’s (Appendix A), and the *ADCYAP1R1* gene was enriched with neuroactive ligand–receptor interaction pathway (Appendix A).

## 3. Discussion

FTD is a neurodegenerative disease affecting behavior and language and connects to the pathology in the brain’s frontal and temporal lobes [13]. FTD patients show a decline in survival rates and show comorbidities with several other diseases. The correlation of the diseases, such as COVID-19 and Breast cancer, to FTD and survival at the molecular level is unknown [14]. So far, there is no known cause for FTD and no actual therapies for the treatment of FTD patients.

Additionally, Breast cancer affects 230,480 women in the USA alone, with 39,520 of these women reaching mortality [15]. These statistical factors are important for our study as we measured the comorbidity of FTD and Breast cancer to find the possible overlap between these two diseases. Valosin-containing protein (*VCP*) is known to be a prognostic biomarker in breast carcinoma [16], and mutations in *VCP* are known to show clinical phenotype for FTD. In this study, we aim to identify such genes which show a role in both Breast cancer and FTD. Furthermore, the development of new diagnostic and prognostic genetic biomarkers for patients with FTD is urgently needed. Differentially expressed genes among racial/ethnic groups have been extensively explored in recent years and may harbor some of the new diagnostic and prognostic genetic biomarkers for FTD. To this end, we performed DEG analysis for the FTD among AA and white patients, and the sample size of these participants is limited. We found 305 DEGs to be significant at a *p*-value < 0.05 (Appendix A). We then analyzed their potential to be involved in autophagy and found the genes *MAPK9*, *AKT3*, *GFAP*, *UBE2N*, and *VDAC1* were involved in autophagy (Figure 1). We then validated whether these five genes have molecular underpinnings of COVID-19-specific comorbidities and identified two genes, *GFAP* and *RTN4*, involved in COVID-19 and FTD (Appendix A). SARS-CoV-2, an etiological agent of COVID-19, is found to occupy the *RTN4* gene, thereby enhancing the development of virally induced double-membrane vesicles, which is crucial for the replication of the viral genome. Increased concentrations of *GFAP* in patients with COVID-19 are found to increase the mortality risk and can be used as a possible biomarker for COVID-19 severity [17]. Then, we analyzed whether any of these FTD DEGs play a key role in Breast cancer prognosis and survival. We found that the FTD DEGs, *AKT3*, *UBE2N*, *VDAC1*, *ADCYAP1R1*, *C4A*, and *GFAP* showed significant comorbidity in breast invasive carcinoma patients at a *p*-value < 0.05 (Figure 2). We analyzed immunotherapeutic drugs interactions (Appendix A), RBP interactions (Appendix A), pathogenicity (Appendix A), genetic alterations (Figure 5), clinical association (Appendix A), and pathway enrichment (Appendix A) of these five novel FTD-Breast cancer comorbid genes. Our findings from these analyses suggested that these five FTD-Breast cancer comorbid genes were novel and play a crucial role in both FTD and Breast cancer.

Overall, we found two novel FTD-COVID-19 comorbid genes, *GFAP* and *RTN4*, and five novel FTD-Breast cancer comorbid genes, *AKT3*, *GFAP*, *ADCYAP1R1*, *VDAC1*, and *C4A*, in this study. The *AKT3* gene plays a role in AKT-kinase, which correlates with serine and threonine stressors. In relation to FTD, this causes deterioration in keeping balance and destroys DNA [18]. The *VDAC1* gene interacts with IP3 receptors that work with sigma-1 receptors, and mutated *SOD1* acts upon these receptors. The sigma-1 receptor disrupts the IP3 receptor pathway of taking Ca^2+^ from the endoplasmic reticulum to the mitochondria, and this disturbance from the sigma-1 receptor causes the mutation of *SOD1* to take place and impair the endoplasmic reticulum mitochondria signaling [19]. *GFAP* is known to correlate with the process of FTD, called astrogliosis, which is the process of trying to amend neurological damage to the brain [20]. The *C4A* gene is involved in inflammatory processes associated with Alzheimer’s and plays a role in the pathology of FTD. Several studies, such as mice and human postmortem, have shown the up-regulation of FTD. The *C4A* gene works in changing the structure and the number of copies in genetic biomarkers. In response to the exposure of *C4A* gene expression, the nuclear ribonucleoprotein gives a transactive response to DNA protein binding [21]. The *ADCYAP1R1* gene is involved in the regulation of adrenocorticotropin and catecholamine hormones. Depletion of catecholamine compounds is linked to various underlying neurological health complications, including loss of or impaired motor function and distinct behavioral changes [22] which are consistent in FTD phenotypes. Up-regulation of the *ADCYAP1R1* gene is likely to influence the pathology of FTD by causing a disruption of adrenocorticotropin and catecholamine levels in the body.

Collectively, the above findings demonstrated that alterations of the FTD-COVID-19 and FTD-Breast cancer comorbid genes, *AKT3*, *GFAP*, *ADCYAP1R1*, *VDAC1*, and *C4A*, were associated with the pathogenesis of FTD, COVID-19, and Breast cancer. These genes could be utilized as potential therapeutic targets for the treatment of comorbidities among these three diseases.

Limitations of our study: our study has a few limitations, and some of these limitations are as follows. There are only a limited number of samples, with eight samples per group. Our study is also limited due to the number of samples from African American backgrounds being very low. We will address these limitations in our future studies by including more datasets and performing RNA sequencing experiments on a comparable number of samples from different ethnic groups for both normal and diseased conditions. In our future studies, we aim to combine different datasets and perform meta-analysis to identify the genes that could serve as possible biomarkers for FTD.

## 4. Materials and Methods

### 4.1. Data Collection and Identification of Differentially Expressed Genes in FTD

The Gene Expression Omnibus (GEO) dataset (GSE193391) from the GEO database [23] with AA and white ethnic background (Table 1) was selected for our study. The samples selected from this study are from the human dorsolateral prefrontal cortex of postmortem brain tissue [24]. These samples have an *APOE* status, which plays a key role in COVID-19 outcomes by down-regulating the *ACE2* and misbalancing the RAS pathway [25]. For the downloaded dataset, we performed differential gene expression (DEG) analysis using the GEO2R tool available on the GEO database. For the identification of differentially expressed genes, log2(fold change [FC]) value > 2, *p*-value < 0.05, and Bonferroni false discovery rate (FDR) < 0.05 parameters were considered as the cut-off threshold values in our study. Box plots between the samples from the control and FTD groups were created using the GEO2R (Appendix A).

### 4.2. Identification of Autophagy Genes

Autophagy is a lysosomal degradation pathway required for survival, differentiation, development, and homeostasis and is known to play a crucial role in diverse pathologies, such as cancer, infections, aging, and neurodegeneration. To understand the autophagy genes in the identified DEGs, we submitted the DEGs that are significant between the healthy controls and the FTD samples to the Human Autophagy Databases (http://www.autophagy.lu/index.html; http://www.tanpaku.org/autophagy/) (accessed on 9 November 2022) [26].

### 4.3. Identification of COVID-19-Associated Genes

To identify the DEGs that are comorbid between FTD and COVID-19, we submitted the DEGs that are significant between the healthy controls and the FTD samples to DisGeNET (http://www.disgenet.org/). DisGeNET is a discovery platform that integrates data from curated repositories of experts, GWAS catalogs, animal models, and their respective scientific literature. It contains 1,134,942 gene–disease associations (GDAs) between various genes, diseases, disorders, traits, clinical or abnormal human phenotypes, and their respective variant associations [27]. DisGNET measures the Disease Pleiotropy Index (DPI), Disease Specificity Index (DSI), and the probability of being loss-of-function intolerant (pLI). DPI is measured by two variables: the value of different disease classes for a disease of interest and the total number of disease classes in DisGeNET, which is 29. DPI ranges from 0 to 1. DSI is the variable that measures the gene or variant association with a disease. DSI ranges from 0 to 1 and has an inversely proportional relationship to the disease number in accordance with the specified gene. pLI, on the other hand, measures how tolerant or intolerant a gene is for the protein truncating variation (variation of nonsense, splice acceptor/donor). A pLI value close to 1 means the gene is intolerant.

### 4.4. Verification of FTD DEGs in Breast Cancer

To verify the expression of identified FTD DEGs with statistically significant differences in Breast cancer based on a large sample size, we performed analysis using GEPIA [28] (http://gepia.cancer-pku.cn/) (accessed on 13 November 2022). For the genes that were significantly expressed in Breast cancer, we used the KM-plotter [29] (https://kmplot.com/analysis/) (accessed on 16 November 2022) to analyze the influence of genes on Breast cancer survival under low and high expression.

### 4.5. Identification of the Alteration in the DEGs

To analyze the alterations in the identified DEGs on large-scale cancer genome datasets, we submitted them to cBioPortal (http://cbioportal.org) (accessed on 13 November 2022), a web server for integrative analysis of complex cancer genomics and clinical profiles (http://cbioportal.org) (accessed on 18 November 2022) [30]. The Genomic Alteration Types and putative copy-number alterations were downloaded for each of the identified DEGs.

### 4.6. Expression Analysis of FTD-Breast Cancer Comorbid Genes

To integrate, analyze, and visualize the expression of public genomic data with the FTD-Breast cancer comorbid genes, we used the UCSC Xena server [31] (https://xena.ucsc.edu/welcome-to-ucsc-xena/) (accessed on 20 November 2022). Xena browser allows us to explore the functional genomic datasets to perform the correlations between variables related to genotype or phenotype. To determine the differences in FTD-Breast cancer comorbid gene expression between tumor and normal tissues, we performed the clustering analysis using heatmaps to compare their gene expression, exon expression, and DNA methylation.

### 4.7. Identification of RNA Interactions for the FTD-Breast Cancer Comorbid Genes

To identify and analyze respective RNA Interactomes for the FTD-Breast cancer comorbid genes, we used ENCORI (The Encyclopedia of RNA Interactomes) (https://rnasysu.com/encori/index.php) (accessed on 24 November 2022), a database with millions of RBP–RNA, RNA–RNA interactions, functions, and mechanisms in human diseases assessed through CLIP-seq and various high-throughput sequencing data [32].

### 4.8. Single-Cell Transcriptome Profiling of FTD-Breast Cancer Comorbid Genes

To identify the transcriptome profile of FTD-Breast cancer comorbid genes at the single-cell level, we have submitted the genes to a single-cell atlas (http://bcatlas.tigem.it) (accessed on 1 December 2022). Single-cell atlas is a web-based server consisting of 35,276 individual cells from 32 Breast cancer cell lines clustered according to either genomic variants or copy number variations. It allows us to study tumor heterogeneity and drug response of Breast cancer cell lines at a single-cell level [33].

### 4.9. Prediction of Potential Therapeutic Drugs

Identification of potential immunotherapeutic drugs for the genes will provide clues for drug development and can be translated into clinical applications. To identify interaction information about the approved and immunotherapeutic drugs for the FTD-Breast cancer comorbid genes, we used the drug–gene interaction database (DGIdb, https://www.dgidb.org/search_interactions) (accessed on 8 December 2022) —a web resource that provides information on drug–gene interactions, databases, druggable genes, and other web-based sources [34]. We selected the drugs with an interaction score greater than one as the cut-off criterion [35].

### 4.10. Pathogenicity Gene Score Analysis

For a given variant, pathogenicity allows us to check if it increases an individual’s susceptibility or predisposition to a certain disease or disorder. Therefore, for each of the identified DEGs, we performed tolerance or pathogenicity gene score analysis using the Vanno Portal (http://www.mulinlab.org/vportal) (accessed on 12 December 2022). Respective pathogenicity scores were taken from various tools such as Combined Annotation Dependent Deletion (CADD), Deep and Neural Network (DANN), base-wise annotation (regBase PAT), spectral approach for coding and non-coding genetic variants (Eigen), finds point mutation (FATHMM-MKL), simpler meta score based on direct, known genes (Eigen_PC), Regulatory Mendelian Mutation score (ReMM), fitness consequence (fitCons), whole genome functional annotation (GenoCanyon), linear model for functional genomic data with probabilistic model (LINSIGHT), and Context-Dependent Tolerance Score (CDTS) in the Vanno portal.

### 4.11. Interpretation of Genetic Variants

To identify phenotypic variation influenced by DEGs, we conducted a genetic variant analysis using 3′aQTL-atlas (https://wlcb.oit.uci.edu/3aQTLatlas) (accessed on 16 December 2022) [10] search by gene/SNP across the human brain frontal cortex tissue. The resulting variants of the queried genes were refined for association with changes in untranslated regions or 3′UTR alternative polyadenylation (APA) site usage based on statistical significance *p*-value < 0.05. Boxplots depicting the allele frequencies of the common genetic variants were extracted from the 3′aQTL-atlas browser to determine genotypic influence and functional interpretation of genetic variations of the genes.

### 4.12. Enrichment of Functional Pathways

To perform functional and pathway enrichment analysis, we have submitted the FTD-Breast cancer comorbid genes to the Database for Annotation, Visualization, and Integrated Discovery (DAVID, https://david.ncifcrf.gov/) (accessed on 6 January 2023) [36]. DAVID offers systematic and integrative functional annotation for researchers to unravel the biological meaning of the submitted genes. Using DAVID, gene ontology (GO) annotation—including the biological process (BP), molecular function (MF), and cellular component—and Kyoto Encyclopedia of Genes and Genomes (KEGG) pathway enrichment analysis [12] were performed on the selected genes.

### 4.13. Statistical Analysis

For comparing the group control and FTD using the GEO2R, we used Benjamini and Hochberg (False discovery rate) for the adjustment of *p*-values. All the statistical procedures provided in the study were performed at a significance level cut-off of *p*-value < 0.05. For comparing the expression genes, we considered the log2 fold change value. For KEGG enrichment analysis, the genes that were significant at a *p*-value < 0.05 were considered as a threshold. For the analysis using GEPIA, we used the following parameters: log2 fold change cut-off 1, *p*-value cut-off 0.05, log-scale log2(TPM + 1), and Jitter size 0.4. For KM-plotter, we used multiple testing correction tools that included the statistical tests Bonferroni, the Holm (step-down), and the Hochberg (step-up) corrections, and allowed the calculation of the False Discovery Rate (FDR) and *q*-values.

## 5. Conclusions

In conclusion, the genes *AKT3*, *VDAC1*, *ADCYAP1R1*, *C4A*, and *GFAP* play a crucial role in the pathogenesis and clinical association of FTD. These genes could be used as novel diagnostic and prognostic biomarkers and therapeutic targets for patients with FTD.

## Figures and Tables

**Figure 1 ijms-24-14910-f001:**
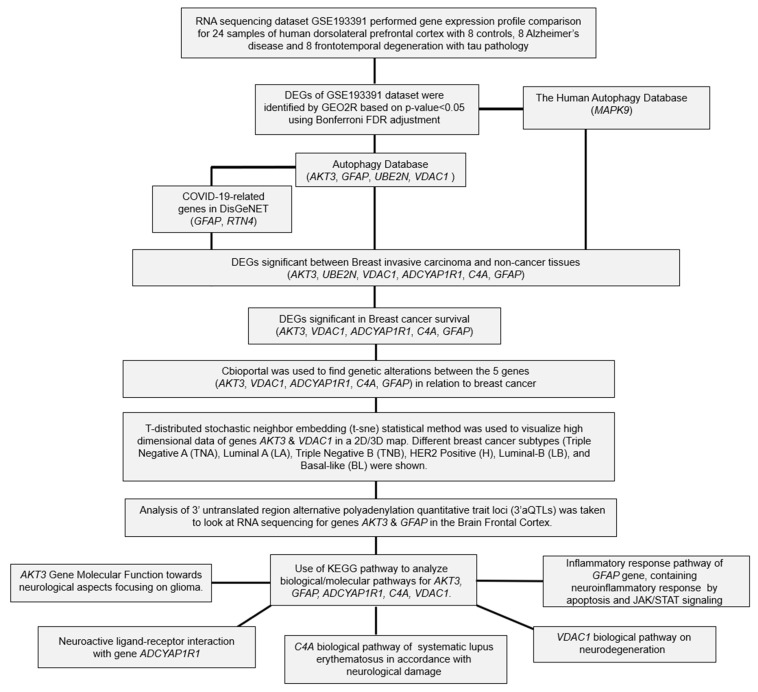
Flow chart of the bioinformatics analysis performed in this study.

**Figure 2 ijms-24-14910-f002:**
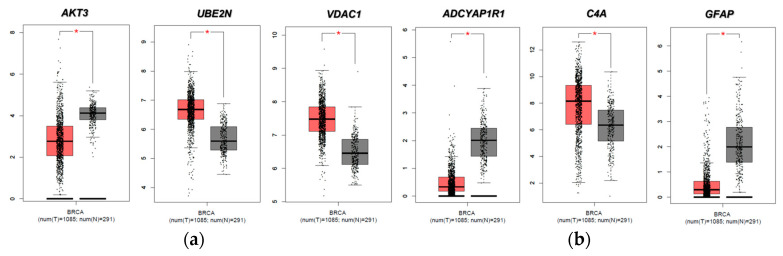
Expression of FTD DEGs in breast invasive carcinoma and non-cancer tissues. (**a**) Breast invasive carcinoma (BRCA) expression profiles of the genes that are up-regulated in FTD. (**b**) Breast invasive carcinoma (BRCA) expression profiles of the genes that are down-regulated in FTD. Red color boxes represent cancer, whereas black color boxes represent normal, and * represents significance at a *p*-value < 0.05.

**Figure 3 ijms-24-14910-f003:**
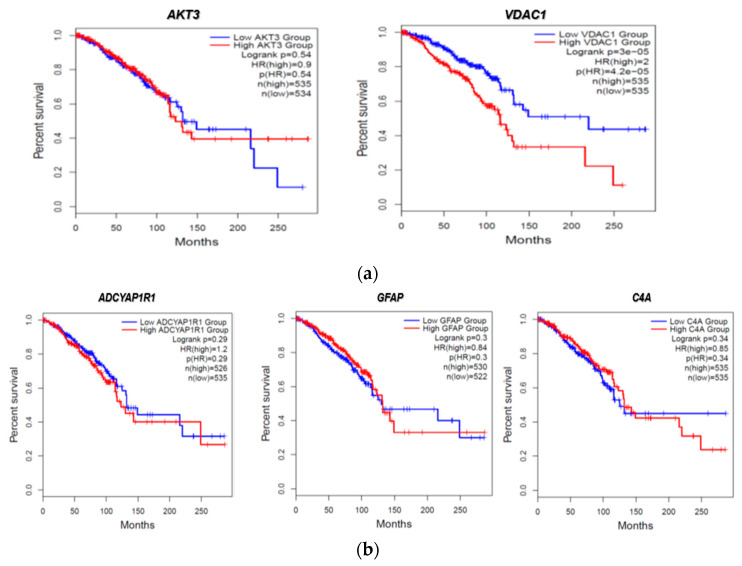
FTD DEGs significantly affecting the overall survival of patients with breast invasive carcinoma (**a**) Up-regulated genes. (**b**) Down-regulated genes.

**Figure 4 ijms-24-14910-f004:**
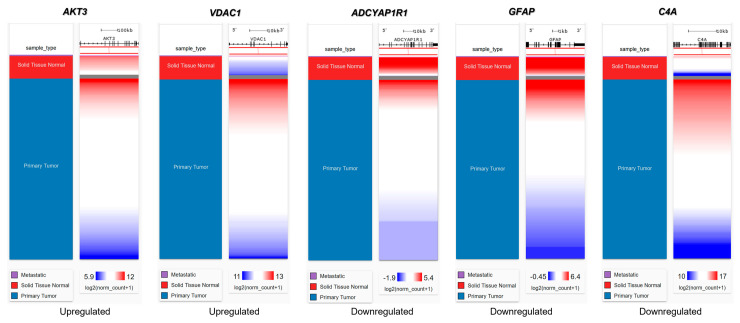
Heat map of FTD-Breast cancer comorbid genes expression in the metastatic, solid tissue normal and primary tumors of TCGA datasets. Metastatic is shown in purple, Solid Tissue Normal in shown in red, and Primary Tumor is shown in blue. The significance of the DEGs was measured in log2(normal count + 1), representing red as the stronger the correlation and blue as the weaker the correlation.

**Figure 5 ijms-24-14910-f005:**
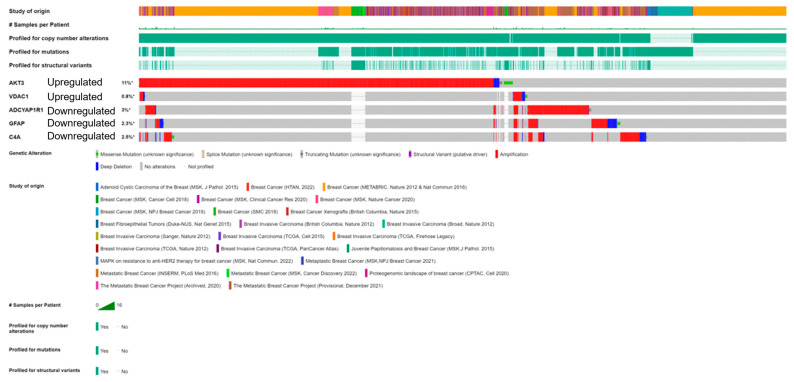
cBioPortal “oncoprint” representation of alterations in FTD genes identified in Breast cancer consisting of 31,500 samples (amplification, red; deletion, blue; mutation, green). Numbers represent the combined frequency of all alterations. TCGA datasets selected are shown above.

**Table 1 ijms-24-14910-t001:** Samples from the GEO dataset GSE193391 that were considered for our study.

Control Samples				
Accession	Age	Sex	Race	APOE Status
GSM5799484	52	Female	White	E3/4
GSM5799495	59	Male	Black	E2/3
GSM5799498	57	Male	White	E3/3
GSM5799485	78	Female	White	E3/3
GSM5799504	56	Male	White	NA
GSM5799489	70	Male	Black	E3/3
GSM5799493	75	Female	White	E3/3
GSM5799506	74	Female	White	E3/3
**FTD Samples**				
**Accession**	**Age**	**Sex**	**Race**	**APOE Status**
GSM5799499	81	Male	White	E3/3
GSM5799500	60	Female	White	E3/4
GSM5799501	64	Male	White	E3/4
GSM5799488	71	Male	White	NA
GSM5799505	56	Female	White	NA
GSM5799490	65	Female	White	E2/3
GSM5799492	65	Female	White	NA
GSM5799494	63	Male	White	E3/3

## Data Availability

Publicly available datasets were analyzed in this study.

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
