# Peer review of "Analysis of Potential Biomarkers in Frontal Temporal Dementia: A Bioinformatics Approach"

_ijms, 2023, doi:10.3390/ijms241914910_

Round 1

Reviewer 1 Report

In the paper “Analysis of Potential Biomarkers in Frontal Temporal Dementia: A Bioinformatics Approach”, Momin and others identified potential biomarkers or susceptible genes for FTD that show comorbidities with diseases such as COVID-19 and breast cancer through bioinformatics analysis. There are conceptual problems with the study and design of the experiments. And also the motivation behind performing different analyses is missing. My specific comments are as follows:

1.       The authors stated multiple times throughout the manuscript that the aim is to study the molecular comorbidity features of FTD and other diseases by comparing the American African and White ethnic group RNA-seq samples. However, the GSE193391 data they used for this work contains only two AA samples and those also belong to the control group. In order to properly support their claim regarding the racial/ethnicity factor, they need a dataset having a comparable number of samples from different ethnic groups for both normal and diseased conditions.

2.       For the comorbidity of FTD and COVID-19, the authors cited Ng et al. “Frontotemporal dementia and COVID‐19: Hypothesis generation and roadmap for future research”. From this cited paper, it is not clear to me how FTD and COVID-19 are related based on molecular features, as the paper mostly talks about how the pandemic affects the behavior and social cognition of  FTD patients.

3.       It is not clear from the manuscript why the authors specifically decided to focus on breast cancer instead of another type of cancer. The paper they cited for the comorbidity of FTD and breast cancer, Kasper et al. doesn’t specifically focus on breast cancer. Moreover, the GSE193391 dataset, the authors used for their analysis contains multiple male samples for both conditions. The authors could have chosen another type of cancer which is not sex-specific. If they really one to focus on sex-specific cancer such as breast cancer, they need to control for gender in their bioinformatics analysis.

4.       The authors must perform an extensive literature survey for the comorbidity.

5.       Line 71: “… significantly expressed between AA and white population, …”. I think it would be “… significantly expressed across AA and white population, …”.

6.       Line 75: “Gene AKT3 showed a cluster of 540 …”. Are these 540 genes? The authors should add that.

7.       The authors should discuss more on how GFAP and RTN4 could be involved with COVID-19.

8.       The authors mentioned that AKT3 and C4A get upregulated and downregulated respectively in the breast cancer patient data. However, Fig. 2a and 2b show the completely opposite trend. This error needs to be corrected.

9.       Writing the name of all the drugs interacting with different genes in the main manuscript is unnecessary. The authors mentioned it in the supp. table and that is fine.

10.   Overall, the problem I have noticed with the paper is the authors performed multiple bioinformatics analyses and for many of them the results were analyzed in a quantitative fashion instead of a qualitative way. The authors need to spend more time explaining the requirement of each of the analyses and explaining their results in detail. 

I don't have any such problem with their English.

Reviewer 2 Report

Introduction:

Please clearly state the hypothesis. 

The Introduction is not really clear, superficial, and do not provide a clear rational about the study and why it is important to study that aspect. 

Results: 

2.1

Please explain DEGs, I do not see it before in the text. The first time I see it it's in the Materials and Methods at the end of the manuscript. 

Please explain what is the rational of using n=8, what are the inclusion and exclusion criteria? 

Why don't you add the table 1 which is critical for understanding the population you studied in the results of the main paper. Why APOE is included as a criteria, please provide a rational for that. 

Rest of the results: What is the rational for studying breast cancer? 

figure 2 and 3 are hard to follow, it is not clear who is high and low AKT3, you discuss immunotherapies, once again where is the rational with that? Figure 4-5 same comment.

If I go by the title of the article "Analysis of potential biomarkers in Frontal Temporal Dementia: A Bioinformatics approach" I don't believe that using breast cancer or COVID to find biomarker in FTD was the best approach, these genes can give you a hint maybe, but due to the complexity and the different pathways involved in COVID, FTD and Breast cancer, at some point some pathways like Adenylate cyclase, the complement, and VDCA1. 

On table 1 in supp data, you mention male and female, did you include males in the breast cancer section?

My recommendation would be to rewrite the MS and provide more compelling data and change the title. The figures are hard to understand, there is no rational, no hypothesis. And I'm not convinced by the data. The inflammatory pathways are de facto involved in these diseases, it is not surprising to have a variation in the expression level. 

I will recommend major revision. 

some grammatical problems here and there

Reviewer 3 Report

Dear authors

This is a very nice manuscript addressing an important medical question. While the sample size is not large, the work is based on previously published data and the authors cannot do much about this. The authors should address this limitation in their discussion. 

Specific comments:

1. GFAP is an exciting finding. In one published paper GFAP reduction was associated with the WFS1 gene (PMID: 35389045) and in the same study, the WFS1 was found to be reduced in FTND samples. Maybe authors can address this question a bit more as this is very relevant to their main finding regarding autophagy.

2. The size limitations should also be discussed. What can be done in the future? Combination of different studies and meta-analysis?

3. What was the tissue for the analysis? I found that it is the brain, but the authors should state this very clearly. Also, how brain transcriptome can be used as a biomarker? The brain is not accessible tissue for diagnostic purposes. Would it be the blood transcriptome then? The authors should discuss this point as well.

Reviewer 4 Report

The authors of the manuscript (MS) entitled „ Analysis of Potential Biomarkers in Frontal Temporal Dementia: A Bioinformatics Approach” present results of bioinformatic investigation of gene expression associated with frontal temporal dementia (FTD), based on analysis of GSE193391 dataset from Gene Expression Omnibus (GEO) database and several other databases .

The authors determined altered expression of several genes in FTD, analysed comorbidity of FTD with breast cancer and COVID-19 patients, based on premises that cancer patients are less likely to develop FTD and COVID-19 worsens FTD symptoms, and based on these results conducted analysis of potential drug interactions. The main finding of the study is indication of potential genes, including AKT3 kinase, as potential therapeutic targets.

The study has clinical relevance, and the results argue for further verification in more targeted and prospective observations in larger population of patients. In general, the MS is well written and concise. However, I have some suggestions/points:

1. There are only 8 patients per group (controls, FTD, and Alzheimer disease patients). Please, clarify that this is the limitation of the dataset used for the study.

2. The dataset contains Alzheimer disease patients. Please, clearly state in the MS if these patients were also included in the analysed or not. If no, please, explain why. I yes, please, provide more information in the MS.

3. Please, add a section on statistical analysis of the data.

4. I suggest referencing another study from the GSE dataset, which showed increased transcription of genes related to astrocytes:

Johnson AG, Webster JA, Hales CM. Glial profiling of human tauopathy brain demonstrates enrichment of astrocytic transcripts in tau-related frontotemporal degeneration. Neurobiol Aging. 2022 Apr;112:55-73. doi: 10.1016/j.neurobiolaging.2021.12.005. Epub 2021 Dec 24. PMID: 35051675; PMCID: PMC8976718.

5. I suggest adding the information on the merit to look for comorbidities with breast cancer and COVID-19 in the abstract, as lack of this information may be confusion for people who only read the abstract.

6. Breast Cancer, breast cancer, Breast cancer – please, unify style throughout the MS.

Round 2

Reviewer 1 Report

It is still not clear to me why AKT3 is mentioned as an upregulated gene in breast cancer samples in the text (lines 116, 117, 120, and 121). Because the Fig 2a (extreme left) box plot shows it has lower expression in breast cancer cells compared to normal. Similarly, for C4A, Fig 2b shows that its expression is higher in cancer cells compared to normal. However, in the text, it is mentioned as a downregulated gene in breast cancer.   

Author Response

We agree with the reviewer’s concern. We have added a few lines in the manuscript and in the figure legends so that the readers get a clear idea.

Reviewer 2 Report

Thanks

Author Response

You are welcome

Reviewer 4 Report

Most of my questions/suggestions have been addressed by the authors.

I suggest to provide more details on statistical analysis.

Author Response

We agree with the reviewer’s suggestion and provided more details on statistical analysis in the manuscript.

Round 3

Reviewer 1 Report

The authors addressed all of my concerns.